# Feasibility of Utilizing Stable-Isotope Dimethyl Labeling in Liquid Chromatography–Tandem Mass Spectrometry-Based Determination for Food Allergens—Case of Kiwifruit

**DOI:** 10.3390/molecules24101920

**Published:** 2019-05-18

**Authors:** Yi-Chen Shih, Jhih-Ting Hsiao, Fuu Sheu

**Affiliations:** Department of Horticulture and Landscape Architecture, National Taiwan University, Taipei 10673, Taiwan; r05628202@ntu.edu.tw (Y.-C.S.); b04608052@ntu.edu.tw (J.-T.H.)

**Keywords:** stable-isotope dimethyl labeling, liquid chromatography–tandem mass spectrometry, food allergen, kiwifruit

## Abstract

Stable-isotope dimethyl labeling is a highly reactive and cost-effective derivatization procedure that could be utilized in proteomics analysis. In this study, a liquid chromatography– tandem mass spectrometry in multiple reaction monitoring mode (LC-MS-MRM) platform for the quantification of kiwi allergens was first developed using this strategy. Three signature peptides for target allergens Act d 1, Act d 5, and Act d 11 were determined and were derivatized with normal and deuterated formaldehyde as external calibrants and internal standards, respectively. The results showed that sample preparation with the phenol method provided comprehensive protein populations. Recoveries at four different levels ranging from 72.5–109.3% were achieved for the H-labeled signature peptides of Act d 1 (SPA1-H) and Act d 5 (SPA5-H) with precision ranging from 1.86–9.92%. The limit of quantification (LOQ) was set at 8 pg mL^−1^ for SPA1-H and at 8 ng mL^−1^ for SPA5-H. The developed procedure was utilized to analyze seven kinds of hand-made kiwi foods containing 0.0175–0.0515 mg g^−1^ of Act d 1 and 0.0252–0.0556 mg g^−1^ of Act d 5. This study extended the applicability of stable-isotope dimethyl labeling to the economical and precise determination of food allergens and peptides.

## 1. Introduction

Sensitive and reliable methods for the determination of food allergens are necessary to detect the undeclared allergens resulting from contamination during food production and to enforce the regulation of allergen labeling [1]. For these purposes, various techniques have been applied to detect either the allergen protein itself or a marker indicating its presence [2]. Enzyme-linked immunosorbent assay (ELISA) is commonly used for allergen analysis. However, ELISA is limited by the possibility of cross-reactivity which leads to false-positive results in this antibody-based assay. DNA-based methods, which are carried out by means of detecting specific DNA fragments instead of the allergenic compounds, could not provide the exact concentration of the allergens [1]. Because of these limitations, alternative strategies for allergen analysis are required. Mass spectrometry (MS)-based proteomic methods, providing high sensitivity and allowing the identification and quantification of allergenic protein, thus have been of great importance for allergen detection [3].

In general, the methodology of MS-based proteomics absolute quantification is to analyze a signature peptide using a synthetic analogue with a stable isotope label as the internal standard [4]. A signature peptide is a surrogate peptide that is unique to the target allergenic protein and fulfills certain criteria, including the length of the peptide, the accuracy of digestion, and the absence of post-translational modification [5]. The stable isotope labels can be introduced into the signature peptide metabolically, enzymatically, chemically, or provided by synthetic peptide standard [6]. However, many of these proposed methods suffer from several disadvantages, such as limited biological applicability for labeling in cell culture, the requirement of a certain amino acid for some chemical labeling methods, and the high cost of isobaric tags, which restrict the applicability in routine use for food samples [7].

Stable-isotope dimethyl labeling, as a chemical labeling method for quantitative proteomics, is achieved by the dimethylation of a primary amine (the N-terminus and side chain of lysine residue) using different isotope forms of formaldehyde and sodium cyanoborohydride (NaBH_3_CN) [8]. The derivatization carried out with deuterated formaldehyde (CD_2_O) produces a mass increase of 32 Da for each reductive site (Figure 1). On the other hand, dimethylation performed with normal formaldehyde (CH_2_O) generates a mass shift of +28 Da per primary amide. In addition, several combinations of reagents with different isotope forms were proposed to produce various mass differences [9]. The reaction of dimethyl labeling completes in minutes without any side product, and this method is much more economical in cost than other developed methods [9,10]. In addition, dimethyl labeling can be practiced on protein from any species [7]. With these advantages, stable-isotope dimethyl labeling has been applied in many quantitative proteomic studies [11,12,13,14]. Nonetheless, to our knowledge, isotope dimethyl labeling has not been applied in any analysis method for either allergenic proteins or food samples.

Kiwifruit (*Actinidia deliciosa*) is currently considered to be one of the common allergenic foods. According to the World Health Organization and International Union of Immunological Societies Allergen Nomenclature Sub-committee [15], 13 kiwifruit allergenic proteins have been officially registered. Actinidin (Act d 1), a thiol-protease, is the major allergen of kiwifruit [16]. Act d 2 belongs to the thaumatin-like protein family which includes several fruit allergens [17]. Act d 3 is a glycoprotein, but its function is still unknown [18]. Act d 4 is characterized as phytocystatin, which is a cysteine protease [19]. Kiwellin (Act d 5), a major protein component of kiwifruit, is a cell-wall-related protein [20]. Act d 6 and Act d 7 are a pectin methylesterase inhibitor and pectin methylesterase, respectively [21]. Both Act d 8, a Bet v 1 homologue, and Act d 9, profilin, are pollen-related allergens [21,22]. Act d 10 is a lipid transfer protein, and Act d 11 belongs to ripening-related protein family [23,24]. Act d 12, an 11S globulin, and Act d 13, a 2S albumin, are two novel allergens that are located in the seeds [25]. With several kiwi allergens being identified, some methods for the analysis of them have been proposed. The quantification of Act d 1 and Act d 2 was carried out by means of ELISA and liquid chromatography–ultraviolet (LC-UV) method, respectively [26,27]. However, these methods suffer from some disadvantages such as false-positive results caused by cross-reactivity in ELISA and overestimation of allergen contents resulting from the low selectivity in absorbance assay [28]. Allergen detection using liquid chromatography–tandem mass spectrometry in multiple reaction monitoring mode (LC-MS-MRM) is performed by analyzing the *m*/*z* values and ion transitions of signature peptides for target allergens, which could improve the selectivity and prevent erroneous quantification [28]. Moreover, higher sensitivity could be achieved within the MRM mode [5], indicating that the LC-MS-MRM could be an appropriate method for the analysis of kiwi allergens at trace level.

In this work, we intend to propose the first-time application of stable-isotope dimethyl labeling to the trace quantification of fruit allergens using LC-MS-MRM. The identification of allergens was carried out with liquid chromatography/electrospray ionization-quadrupole-time of flight mass spectrometry (LC/ESI-Q-TOF). The identified allergens with high coverage and MASCOT scores were selected as target allergens. To develop the platform for the quantification of kiwi allergens, the external and internal standards were prepared by isotopically labeling the synthetic signature peptide standards with stable-isotope dimethyl labeling. In addition, some steps for protein extraction described in a previous study [29] were optimized to be more time-efficient. Moreover, the established procedure was validated with certain criteria and was applied to self-made kiwi foods to examine the applicability for foodstuff analysis.

## 2. Results

### 2.1. Evaluation for Protein Extraction Methods

To determine the appropriate extraction method which could produce the most comprehensive protein populations, kiwi proteins yielded by phenol, trichloroacetic acid (TCA), ammonium sulfate, and sodium chloride methods, were analyzed with SDS-PAGE (Figure 2). Four allergenic proteins, namely, actinidin (Act d 1), thaumatin-like protein (TLP) (Act d 2), glycoprotein (Act d 3), and kiwellin (Act d 5) were chosen as the indicators to evaluate the extraction quality of each method. The ammonium sulfate method (Lane 3, Figure 2) which gained only one (Act d 1) of the four indicators showed poor extraction efficacy. On the other hand, the sodium chloride method obtained three of them, while the phenol and TCA methods yielded all the indicators. In particular, both the phenol and TCA methods spanned broadly in the region from 10 kDa to about 70 kDa markers (Lanes 2, 4, and 5, Figure 2), and the amounts of proteins extracted by means of the phenol and TCA methods were 4.47 mg g^−1^ and 4.05 mg g^−1^, respectively, indicating that these two methods could provide the most complete repertoire of proteins with an extraction recovery higher than 4 mg g^−1^. To prevent the extracted proteins from degradation which could lead to recovery loss within the sample preparation steps, the phenol method which could minimize protein degradation [30] was eventually chosen over the TCA method for sample preparation in the following experiments.

### 2.2. Determination of Signature Peptides for Kiwi Allergens

For the establishment of an LC-MS-MRM approach for protein quantification, selection of signature peptides from the tryptic peptides was needed. The protein extracts underwent tryptic digestion and were then analyzed using LC/ESI-Q-ToF. The mass spectral interpretation of proteins in the kiwifruit was performed with the Mascot Distiller, and eight of the kiwifruit allergenic proteins were identified (Appendix A). Among the eight allergenic proteins, Act d 1, Act d 5, and Act d 11 were selected as target allergens for quantification due to high coverage and the MASCOT score in protein analysis. Tryptic peptides verified from these three proteins served as candidate signature peptides (Figure 3). Further screening was carried out in accordance with several criteria: 8–19 amino acids in length to fit the scan range limit of the mass spectrometer, the absence of cysteine and methionine, which could result in chemical modification, the exclusion from internal tryptic cleavage sites, and ragged end to stabilize the accuracy of tryptic digestion [5]. Peptides fulfilling the requirements described above were subjected to a BLAST search to confirm if they were specific to the respective allergens and species (Appendix A). Peptide ^135^SAGAVVDIK^143^, which met all the criteria and was absent in other proteins and species turned out to be the signature peptide for Act d 1 (SPA1). Instead of peptide ^106^IVALSTGWYNGGSR^119^, which existed in other species, peptide ^160^NNIVDGSNAVWSALGLDK^177^ was selected to represent Act d 5 (SPA5). For Act d 11, peptide ^109^GEHNSVTWTFHYEK^122^ was determined as the surrogate peptide (SPA11). The three chosen peptides were artificially synthesized as standards for the establishment of the quantification methods.

### 2.3. Sample Derivatization and Preparation of Internal Standards

An internal standard is another prerequisite for absolute quantification to calibrate the loss and error throughout the whole analysis procedure. For the preparation of internal standards, the signature peptides were reacted with CD_2_O to form a dimethyl group on the N-terminus and a lysine residue side chain. On the other hand, the external calibrants and the samples were derivatized with CH_2_O. To investigate the exact mass shift and the efficiency of the labeling reaction, we analyzed the peptide markers with and without a dimethyl label using a mass spectrometer. As shown in Figure 4, +56 and +64 *m*/*z* shift for the H- and D-labeled SPA1 (SPA1-H, *m*/*z* 915 and SPA1-D, *m*/*z* 923) were observed when compared with SPA1 (*m*/*z* 859). In addition, the ion peak of the unlabeled SPA1 was not detected at the *m*/*z* of 859 in spectra of SPA1-H and SPA1-D, implying the complete labeling of the peptide standards. On the other hand, the respective increase in *m*/*z* values for the H- and D-labeled SPA5 (SPA5-H, *m*/*z* 965 and SPA5-D, *m*/*z* 969) were 28 and 32 (Appendix A). Relative to SPA11 (*m*/*z* 579), there was a +19 and a +21 *m*/*z* shift for the H- and D-labeled SPA11 (SPA11-H, *m*/*z* 598 and SPA11-D, *m*/*z* 600) (Appendix A). The ion peaks of the unlabeled SPA5 were not detected at the *m*/*z* of SPA5 (*m*/*z* 937), nor detected at that of unlabeled SPA11 (*m*/*z* 579). Taken together, these MS results demonstrated that the synthetic peptides were mostly labeled with dimethyl, indicating that stable-isotope dimethyl labeling provided high labeling efficiency and that the H- and D-labeled peptides could be properly produced by stable-isotope dimethyl labeling as external calibrants and internal standards, respectively, for the development of the quantification methods. 

### 2.4. Optimization for LC-MS-MRM Parameters

In this study, LC-MS-MRM assay was applied for protein quantification. To develop the LC-MS-MRM methods, the conditions of both the mass spectrometer and the LC gradients were first optimized to ensure that the parameters could maximize the intensity of the peptides and that the peptides could be separated appropriately with the LC system.

The transitions and optimized parameters of the MRM methods for the following data acquisition are summarized in Table 1. In addition to the optimization of MRM parameters, separation of peptides was accomplished with a linear gradient. The MRM chromatograms for SPA1-H and SPA1-D standards are shown in panels A and B of Figure 5, respectively, and those for the labeled form SPA5 and SPA11 are shown in Appendix A. The retention time of each peptide was obtained using the standards and the results are summarized in Table 1. It should be mentioned that SPA1-D, as the isotope-labeled internal standard, was co-eluted with SPA1-H at almost the same retention time. When it comes to SPA5-D and SPA11-D, the same situation was also observed. These results indicated that the D-labeled peptides, co-eluting with the target analytes, could correct not only the loss during sample preparation but also the instrumental errors in the LC and MS system. 

### 2.5. Method Validation

Prior to the quantification of the allergen commodities, this method was assessed for its linearity, sensitivity, recovery, and repeatability. The calibration curves were generated by plotting the ratio of the area of the external calibrants and internal standards to the concentration of the spiked external calibrants. As shown in Table 2, good linearity (R^2^ ≥0.99) was observed in the range of 0.008–4000 ng mL^−1^ for SPA1-H, in the range of 8–4000 ng mL^−1^ for SPA5-H, and in the range of 20–4000 ng mL^−1^ for SPA11-H. Regarding the sensitivity, the limits of detection (LODs) ranged between 8 pg mL^−1^ and 8 ng mL^−1^, and the limits of quantification (LOQs) were between 8 pg mL^−1^ and 20 ng mL^−1^, depending on the signature peptides.

In addition, the assessment of accuracy and precision was performed with the kiwi matrices spiked with H-labeled standards at five concentration levels and 500 ng mL^−1^ of internal standards. Regarding the validation results of SPA1-H and SPA5-H, the lowest recovery was observed for SPA1-H (65.5%), and the highest coefficient of variation was observed for SPA5-H (17.2%) (Table 2). The range of recoveries was 75.6–109.3% for SPA1-H within the spiking range of 1000–12500 ng and 72.5–86.3% for SPA5-H within the spiking range of 2000–25000 ng, and the coefficients of variation for SPA1-H and SPA5-H were 1.86–9.92% and 3.44–9.55%, respectively, which indicated that good robustness could be achieved when analyzing SPA1-H and SPA5-H. On the other hand, the percentage of SPA1-H recovery was from 39.8% to 341.9% with a coefficient of variation ranging from 11.7% to 72.3% (Table 2). The high variation for SPA11-H observed in the validation assessment indicated that accurate and precise quantification of SPA11-H could not be carried out by means of this analytical workflow, and SPA11-H was excluded in the further analysis of food samples. In contrast, the quantification of SPA1-H and SPA5-H could be performed with this developed procedure.

### 2.6. Analysis of Kiwi Food Samples

The proposed workflow for sample preparation and LC-MS-MRM assay was applied to the analysis of kiwifruit and kiwi foods. All the kiwi-based foods, including kiwi jam, hot-air-dried kiwi, lyophilized kiwi, pasteurized puree, high-pressure-processed (HPP) puree, pasteurized juice, and HPP juice, were self-made to control the amount of kiwifruit content in the foods. In addition, to promote the operation and to reduce the reagents’ use during sample preparation, some steps in the literature [29] were modified for low-volume extraction. The amount of kiwi pulp used in the step of sample preparation was reduced to 0.07 g, the total volume of extraction buffer used was reduced to less than 500 μL, and the reaction time for the protein precipitation step was shortened to 4 h. Furthermore, to investigate the efficacy of the modified method, we performed one phenol extraction plus three additional back extractions. The first three extractions gained approximately 73.7%, 20.2%, and 6.1% of the total extracted protein, respectively (Appendix A). On the other hand, the third back extraction produced little to no protein, which showed that almost all the extractable proteins could be obtained with two additional back extractions. After several modifications, the optimized procedure and the procedure previously described [29] were eventually carried out to investigate the extraction efficiency, and the former was found to achieve mostly the same extraction efficiency (Appendix A). Naturally, the modified procedure which was easy to operate and economical in time was applied to the sample preparation in the following experiments.

Samples prepared with the procedure described above were then analyzed using LC-MS-MRM (panels C and D of Figure 5 and Appendix A). The content of Act d 1 and Act d 5 was quantified with the peak area ratio of the corresponding transition for the derivatized signature peptides. The content of Act d 1 in kiwi products was in the range of 0.0175–0.0515 mg g^−1^ product weight (Panel A of Figure 6). As can be seen, kiwi jam and hot-air-dried kiwi contained less Act d 1 with concentrations of 0.0175 and 0.0319 mg g^−1^, respectively, than fresh fruit and other products with concentrations between 0.0476 and 0.0511 mg g^−1^. Panel B of Figure 6 showed the content of Act d 5 ranged from 0.0252 to 0.0556 mg g^−1^. The lower content of Act d 5 was observed in products with thermal processing (kiwi jam, pasteurized juice, pasteurized puree, and hot-air-dried kiwi) when compared with fresh fruit, which contained 0.0495 mg Act d 5 in one gram of kiwifruit. Moreover, all the products with thermal processing contained less Act d 5 than those without thermal processing, indicating that Act d 5 was a heat intolerant allergen. However, regarding Act d 1, only some products with thermal processing (hot-air-dried kiwi and kiwi jam) presented lower content of Act d 1 when compared with those without thermal processing, showing that Act d 1 was stable during pasteurization and that a long period of hot-air drying and severe thermal process could result in the decrease of the content of Act d 1. 

## 3. Discussion

The sample preparation and protein extraction were the key steps for trace peptide determination. Compared with the ammonium sulfate precipitation and sodium chloride extraction methods, the phenol extraction coupled with methanolic ammonium acetate precipitation produced a more comprehensive population of kiwi proteins in fruit juice samples (Figure 2). This could result from the two-step procedure in the phenol method, in which the first phenol extraction excluded interfering components such as water-soluble sugars and acids in food samples, while the following ammonium acetate precipitation increased the purity of the proteins. On the other hand, the ammonium sulfate and sodium chloride methods were direct precipitations that could obtain more hydrophilic compounds resulting in poor protein quality. In addition, the phenol method could further inhibit protein degradation and retain the protein repertoire as well [30]. Therefore, the phenol method was suggested to extract proteins from the complex matrix and food samples with low protein content.

A signature peptide, the critical element for protein identification and quantification, should fulfill several criteria to guarantee its stability and specificity. It was reported that the peptide containing arginine and asparagine was susceptible to deamidation, which could lead to errors in MS-based quantitative methods [31]. However, in this study, the quantification of SPA5-H, which contains three asparagines was not influenced by deamidation, where high recovery and low coefficient of variaon were observed in the validation assessment for SPA5-H (Table 2). This is probably because SPA5-D, as the chemically analogous internal standard, would show almost the same degree of deamidation as the SPA5-H and thus the variation resulting from deamidation was adjusted. In other studies using an isotope internal standard strategy, good robustness could be achieved as well when analyzing signature peptides containing arginine and asparagine [32,33], which supported our contention that the isotope internal standard could correct the errors caused by deamidation and indicated that a peptide being prone to deamidation could be selected as a signature peptide for protein quantification with the employment of an isotope internal standard.

For accurate and precise quantification, a chemically analogous isotope internal standard was essential to compensate for recovery loss and systematic bias within the analysis procedure. In this study, the artificial synthetic signature peptides were derivatized with CD_2_O as the isotope internal standard. The peptides methylated with CH_2_O served as the standards for calibration curves. It was observed that the dimethylation of the signature peptide standards had been fully completed without any unlabeled peptide detected (Figure 4 and Appendix A), and highly linear calibration curves were established. In addition, all the signature peptides in the kiwi food samples were labeled with CH_2_O as well to minimize possible variation and errors resulting from the incomplete reaction and to correct and maintain the recovery in sample preparation and the derivatization process. Fortunately, there were no unreacted allergen peptides detected within food samples after dimethylation, which indicated that the reaction of food samples was highly effective and efficient. Moreover, the isobaric tag for relative and absolute quantification (iTRAQ) which labels isotopic *N*-methylpiperazine onto peptides is commercially available, and it has been widely utilized for various types of sample matrices such as blood, plant tissues, and food samples [34,35,36]. Unlike the quantification by dimethylation which labeled the precursor ions of H- and D-labeled peptides and their sequence fragment ions, the iTRAQ method was carried out by means of detecting the *N*-methylpiperazine tag as the reporter ion of the peptides. Hence, the efficiency of the labeling reaction would highly influence the number of reporter ions and the recovery of target proteins. The results in which no unlabeled peptides were detected in the MS spectra (Figure 4 and Appendix A) revealed that formaldehyde methylation was an effective and efficient derivatization and thus suggested that dimethyl labeling could be an alternative method to iTRAQ.

To assure the reliability of this study, certain parameters such as precision and accuracy needed be evaluated. In the validation experiments, the H-labeled peptide standards and the D-labeled internal standards were spiked before the trypsin digestion to examine the recovery and reproducibility for sample preparation. After the experiments, good robustness could be observed when analyzing SPA1-H and SPA5-H (Table 2). These results showed that, with respective labeling of CH_2_O and CD_2_O on the analytes and the internal standards, the systematic errors such as recovery loss within the desalting process and inconsistent ionization efficiency for analytes could be adjusted, and accurate and precise quantification could be achieved. Furthermore, compared with other commonly used labeling approaches such as metabolic labeling and the iTRAQ method, dimethyl labeling was more convenient and accessible due to its low cost and ease of operation [9]. Taken together, the dimethyl labeling method could be a recommended strategy for routine analysis.

The analytical procedure was validated for SPA1-H and SPA5-H, however, a lower S/N ratio and a high coefficient of variation were observed when it came to SPA11-H. This result could be attributed to the poor performance of SPA11-H on charge competition during the ionization within ESI and led to poor ionization. As reported previously, the nonpolar residues contributed to the fraction of peptides from ESI droplets, and hydrophobic residues could reduce the ability of peptides in charge competition [37]. Thus, SPA11-H containing only two nonpolar amino acids could have been less successful in competing for charges with the matrix and led to unstable ionization efficiency. To prevent this disadvantage, an amino acid with hydrophilicity was suggested to be taken into consideration in the selection of the signature peptide for quantitative protein analysis using dimethyl labeling coupled with LC/ESI-MS/MS.

The workflow in this study was finally applied to kiwi foods, which were low in protein content. This extended the applicability of stable-isotope dimethyl labeling to the MS-based quantification of kiwi allergens at trace level and revealed the potential of this labeling method in protein analysis of food samples. Unlike blood or urine samples, in which the matrix is normally composed of water, proteins, and electrolytes, the matrix of food samples varies based on the food materials. In fruits and vegetables, there would be more cellulose and low proteins, but carbohydrates, lipids, and proteins would be the major components when it comes to soybean seeds. For meats and seafood, the samples could have high protein and lipid content. Therefore, the sample extraction and preparation steps would be critical for different food samples to remove compounds that could lead to interference in the following sample preparation and instrumental analysis. With an appropriate extraction procedure such as the phenol method in this study that removed the contaminants, a complete derivatization reaction could be achieved which resulted in accurate analysis with the stable-isotope dimethyl labeling method.

In conclusion, we first introduced the stable-isotope dimethyl labeling method to develop a reliable and economical LC-MS/MS quantitative procedure for kiwi allergens. The high efficiency of the dimethyl labeling was proved, and the developed procedure was validated and used on kiwi food samples for trace allergen analysis. This dimethyl labeling strategy could be further utilized and applied to the quantification of allergens in different food matrices and could have potential for the analysis of bioactive peptides and of peptide markers for food authenticity.

## 4. Materials and Methods 

### 4.1. Materials and Reagents

Dithiothreitol (DTT), iodoacetamide (IAA), ethanol, ammonia, formic acid, formaldehyde (37% solution in H_2_O), formaldehyde-*D*_2_ (20% solution in H_2_O), phenol, sodium deoxycholate (SDC), trichloroacetic acid (TCA), ethylenediaminetetraacetic acid (EDTA), β-mercaptoethanol, and triethylammonium bicarbonate buffer (TEAB) were purchased from Sigma-Aldrich (St. Louis, MO, USA). Acrylamide, ammonium persulfate, sodium dodecyl sulfate (SDS), *N*,*N*′-methylene-bis-acrylamide, glycine, and *N*,*N*,*N*′,*N*′-tetraethylmethylenediamine were obtained from Bio-Rad (Hercules, CA, USA). MS grade trypsin was provided by Fisher Scientific (Houston, TX, USA). Acetonitrile (MS grade), acetone, Coomassie brilliant blue, methanol, Tris-Base, and ammonium acetate were from J.T.Baker (Phillipsburg, NJ, USA). Sodium cyanoborohydride was purchased from Alfa Aesar (purity 95%, Haverhill, MA, USA). Urea was from VWR International (Radnor, PA, USA). Ammonium sulfate was obtained from Taiwan Fertilizer Co. (Hsinchu, Taiwan). The peptide standards, IVALSTGWYNGGSR (SPA1), NNIVDGSNAVWSALGLDK (SPA5), and GEHNSVTWTFHYEK (SPA11) were synthesized at the National Institute of Infectious Diseases and Vaccinology (Taipei, Taiwan). Kiwifruits were purchased from a local market in Taipei city.

### 4.2. Phenol Extraction

The phenol extraction method was modified according to a previously described method [29]. Seventy μL of the extraction buffer (1.8 M sucrose, 240 mM Tris-HCl, pH 8.0, 24 mM EDTA, and 0.96% (*v*/*v*) β-mercaptoethanol) were added to 0.07 g peeled and homogenized kiwifruit, and 100 μL of saturated sucrose in phenol was added subsequently. After mixing for a few seconds, the mixtures were placed in an ultrasonicator bath for 20 min at 40 °C and centrifuged at 3900× *g* for 15 min at 25 °C. Afterwards, the supernatants (the phenol phase) were transferred to a new tube, and the lower aqueous phase was extracted twice with the same steps as described above. Proteins were precipitated from the collected phenol phase with five volumes of 0.1 M ammonium acetate in methanol at −20 °C for 4 h and then pelleted by centrifugation at 21,100× *g* for 10 min at −9 °C. The protein pellets were further washed with 0.1 M ammonium acetate in methanol, 80% ice-cold (*v*/*v*) acetone, and cold 70% (*v*/*v*) ethanol. After drying with a vacuum dryer, the protein pellets were finally stored at −20 °C.

### 4.3. TCA Extraction

Kiwifruit was first peeled and then homogenized by the blender. Four grams of the treated kiwi puree was added to 25 mL TCA buffer (10% (*w*/*v*) TCA and 2% (*v*/*v*) *β*-mercaptoethanol in acetone). Extracts were mixed for a few seconds and precipitated for 1 h at −20 °C. The precipitated proteins were pelleted by centrifugation at 5000× *g* for 30 min at 4 °C and washed twice with 10 mL ice-cold acetone. The pellets were dried with a vacuum dryer and resuspended in 15 mL tank buffer (25 mM Tris-base, 0.2 M glycine, and 0.1% (*w*/*v*) SDS). The extracts were centrifuged at 5000× *g* for 10 min at 4 °C, and the resulting supernatants were transferred to a new tube and stored at −20 °C.

### 4.4. Ammonium Sulfate Precipitation

Peeled kiwifruit (250 g) was blended with 250 mL of extraction buffer (100 mM Tris-HCl, 2 mM NaCl, and 10 mM EDTA). After centrifugation (4500 rpm, 15 min, 25 °C), the supernatants were mixed with ammonium sulfate to 95% saturation and precipitated overnight at 4 °C. The precipitated protein was collected by centrifugation (9300× *g*, 15 min, 4 °C). Afterward, the protein pellets were suspended in dialysis buffer (200 mM Tris-HCl, 20 mM EDTA, 2 mM NaCl, 0.8% (*v*/*v*) β-mercaptoethanol) and dialyzed against water for at least 24 h. The treated solution was finally stored at −80 °C.

### 4.5. Sodium Chloride Extraction

This method was carried out as described previously [38]. One mL extraction solution (0.5 M NaCl, pH 8.3, 10 mM DTT) was added to 0.7 g of the peeled and homogenized kiwifruit. Extracts were placed on ice for 1 h and centrifuged at 16,000× *g* for 10 min at 4 °C. The supernatants were transferred to a new tube and stored at −20 °C.

### 4.6. SDS-PAGE Analysis

Extracted proteins were separated via polyacrylamide gel electrophoresis (PAGE) using 12% SDS-PAGE gels at a voltage of 60–130 V. After the separation, the proteins were visualized with Coomassie blue G-250, and the gels were finally digitized with a scanner.

### 4.7. Tryptic Digestion

For samples subjected to LC/ESI-Q-TOF analysis, the protein pellets collected from the phenol extraction were resuspended with 0.5 mL of resuspending buffer (9 M urea in 1 M TEAB buffer). The protein solution was then diluted to a concentration range of 1.5–2.4 μg μL^−1^. Reduction of the disulfide bonds was performed with 0.6 μL of 100 mM DTT, and the alkylation of the cysteine residues was carried out in the dark for 30 min with 0.9 μL of 500 mM IAA. Afterward, 4 μL of 100 mM DTT was added, and the treated solution was incubated for 10 min to quench the alkylation reaction. Prior to tryptic digestion, the solution was first diluted with 182 μL of water. Finally, 2.5 μL of trypsin (0.1 mg mL^−1^) was added to the samples for 16-h digestion at 37 °C.

The digestion protocol for samples for LC-MRM analysis was as follows. The protein pellets from the phenol method were added to 100 μL of resuspension buffer (5% SDC in 50 mM TEAB buffer) and 1 μL of 500 mM DTT in 50 mM TEAB. The protein pellets were then resuspended by pipetting. For the protein solution, disulfide bonds were reacted with DTT for 30 min at room temperature, and the cysteine residues were alkylated with 8 μL of IAA (50 mM in TEAB buffer) for 30 min in the dark at room temperature. The alkylation reaction was quenched with 6 μL of DTT (500 mM in TEAB buffer) for 10 min at room temperature. The treated protein solution was then diluted with 885 μL of TEAB buffer. Afterwards, 35 μL of the diluted protein solution was transferred to a new tube and combined with 15 μL of internal standard solution, formed by mixing the D-labeled peptide standards, SPA1-D, SPA5-D, and SPA11-D, with a concentration of 25 μg mL^−1^ of each in the ratio 1:1:1 (*v*/*v*/*v*). The mixed solution was finally added with 2.5 μL of trypsin (0.1 mg mL^−1^) and incubated for 7 h at 37 °C.

### 4.8. Stable Isotope Dimethyl Labeling

This procedure was modified from the study previously proposed [8]. For the preparation of standards, 5 μL of 4% (*v*/*v*) formaldehyde solution (light, CH_2_O; heavy, CD_2_O) was added to 25 μL of the peptide standards (1 mg mL^−1^) diluted with 100 μL of TEAB buffer. Immediately after the addition of formaldehyde, 5 μL of freshly prepared 0.6 M NaBH_3_CN was added. After incubation for 1 h at 20 °C, the derivatization reaction was quenched by adding 20 μL of 1% (*v*/*v*) ammonia. Then, 10 μL of 5% (*v*/*v*) formic acid was added to acidify the solution. It should be noted that the acidification needs to be performed on ice. After that, the solution of peptide standards was finally diluted to a concentration of 25 μg mL^−1^ with 835 μL of water.

For the preparation of samples for LC-MS-MRM analysis, the dimethylation was carried out after the tryptic digestion. Samples undergoing tryptic digestion were added with 1.5 μL of 4% (*v*/*v*) CH_2_O, and 1.5 μL of freshly prepared 0.85 M sodium cyanoborohydride was then added straight after the addition of formaldehyde. The reaction was quenched with the addition of 6 μL 1% (*v*/*v*) ammonia after 1 h of incubation at 20 °C and was then acidified with 3.5 μL 5% (*v*/*v*) formic acid. The treated samples were finally centrifuged at 15,700× *g* for 2 min to pellet SDC, and 52 μL of the supernatants were transferred to a new tube.

### 4.9. Zip-Tip Desalting

Prior to LC-MS/MS analysis, samples were desalted using the Zip-Tip C18 pipette tips. The tips were pre-wetted with acetonitrile and then equilibrated with 0.1% (*v*/*v*) formic acid before use. The peptides were bound to the tips by approximately 30 times of up-down pipette draws. The tips were then washed with 0.1% (*v*/*v*) formic acid for desalting. After that, the peptides were eluted with 20 μL of 60:40 acetonitrile/0.1% formic acid (*v*/*v*) by aspirating and dispensing the elute solution for about 20 times. The eluted solution was finally diluted with 180 μL of water and stored at −20 °C for further analysis.

### 4.10. LC/ESI-Q-TOF Analysis

Samples treated with digestion and desalting were subjected to LC/ESI-Q-TOF analysis. The peptide mixtures were separated with a Waters Acquity solvent delivery system. Mobile phase A was 0.1% (*v*/*v*) formic acid in water whereas mobile phase B was 0.1% (*v*/*v*) formic acid in acetonitrile. The separation was carried out on an *ACQUITY* UPLC^®^
*Peptide BEH* C18 nanoACQUITY column (1.7 μm, 75 μm × 250 mm) at a stationary flow rate at 0.25 mL min^−1^ with 120 min total run time, and the injection volume was 5 μL. The mobile phase gradient started from 2% to 10% mobile phase B over 0.1 min; mobile phase B was then linearly ramped 40% over 90 min; from 90.1 min to 95 min the amount of mobile phase B increased linearly from 40% to 85% and this proportion was maintained for another 5 min. Following this, mobile phase B was changed to 2% in 5 min, and the column was re-equilibrated under the initial condition for 15 min. The eluted peptides were ionized with positive-ion electrospray ionization and analyzed using a Waters nanoACQUITY-SYNAPT G2 HDMS hybrid quadrupole-time of flight mass spectrometer. The capillary voltage was set at 2.75 kV, sample cone at 40 V, desolvation temperature at 300 °C, and source temperature at 100 °C. Data of full scan and MS/MS scan were obtained in the range of *m*/*z* 350–1700 and of *m*/*z* 50–2000, respectively.

### 4.11. LC-MS-MRM Analysis

The samples for quantification were analyzed using a Waters ACQUITY UPLC-Quattro Primer XE mass spectrometer. The *ACQUITY* UPLC^®^
*Peptide BEH* C18 nanoACQUITY column (1.7 μm, 2.1 mm × 50 mm) was run at a flow rate of 0.25 mL min^−1^. Mobile phase A was 5 mM ammonium acetate in water with 0.1% (*v*/*v*) formic acid and mobile phase B was 0.1% (*v*/*v*) formic acid in acetonitrile. The column was equilibrated using 15% mobile phase B and the samples were injected with 10 μL injection volume. A linear gradient was employed from 15% to 40% mobile phase B over 3 min; mobile phase B was then ramped to 90% from 3 min to 3.25 min and was maintained for 1 min with this proportion. Afterward, mobile phase B was changed to 15% in 0.5 min and was re-equilibrated for 5 min. The mass measurement system was operated in positive mode. Solutions were sprayed through a capillary held at 2.8 kV. The source temperature was set at 120 °C, desolvation temperature at 450 °C. The quadrupoles were scanned in MRM mode with a dwell time of 0.05 s. The cone voltage and collision energy for each transition were derived from repetitive tuning to obtain the highest signal intensity and are summarized in Table 2. Chromatograms and mass spectra were recorded and processed using MassLynx (Waters, v. 4.1, Milford, MA, USA).

### 4.12. Validation Experiments

The analytical method was validated for the three allergens in accordance with the International Conference on Harmonization of Technical Requirements of Registration of Pharmaceuticals for Human Use guidelines Q2 (R1).

The calibration curves were established with diluted peptide standards over a concentration range of 0.008–4000 ng mL^−1^ and internal standards at a concentration of 500 ng mL^−1^. The LOD and LOQ were determined on the basis of sign-to-noise ratio (S/N), where the LOD >3 and the LOQ >10. For the assessment of the accuracy and precision, five different concentration levels of the H-labeled peptide standards were spiked before tryptic digestion was analyzed. The spiked amounts of the quality control standards were 125–12500 ng for SPA1-H, 250–25000 ng for SPA5-H, and 12.5–1250 ng for SPA11-H.

### 4.13. Statistical Analysis

All the error bars of the bar charts were ±SD of at least three independent experiments by performing duplicates. The difference between experimental groups was determined with one-way analysis of variance. A *p*-value under 0.05 was considered a statistically significant difference. 

## Figures and Tables

**Figure 1 molecules-24-01920-f001:**
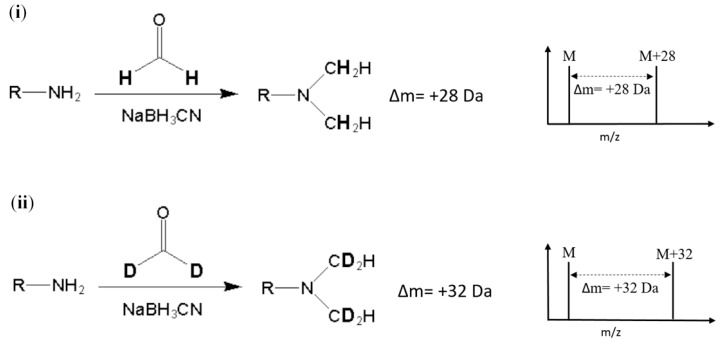
Reaction and *m*/*z* shift of (**i**) light (hydrogen) and (**ii**) heavy (deuterium) stable-isotope dimethyl labeling. R represents the remainder of the peptide and M represents the *m*/*z* of the native peptide with a single charge.

**Figure 2 molecules-24-01920-f002:**
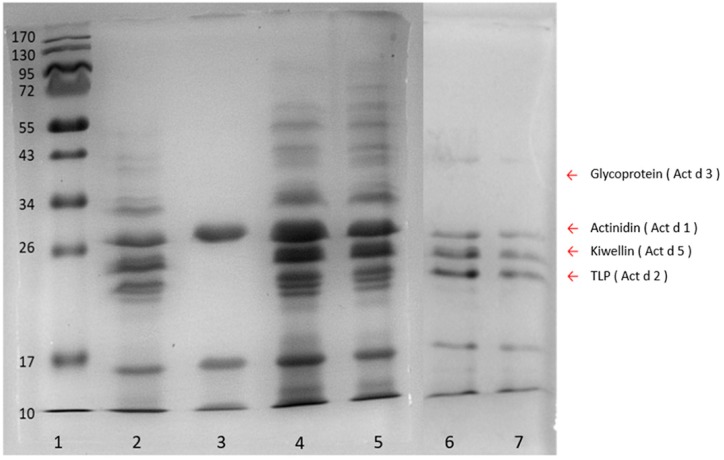
SDS-PAGE of the kiwifruit extracts from four different protein extraction methods. Lane 1, protein ladder; lane 2, phenol method; lane 3, ammonium sulfate method; lanes 4 and 5, TCA method; lanes 6 and 7, sodium chloride method. Arrow indicators on the right indicate four indicator proteins, glycoprotein (Act d 3) (40 kDa), actinidin (Act d 1) (30 kDa), kiwellin (Act d 5) (26 kDa), TLP (thaumatin-like protein) (Act d 2) (24 kDa).

**Figure 3 molecules-24-01920-f003:**
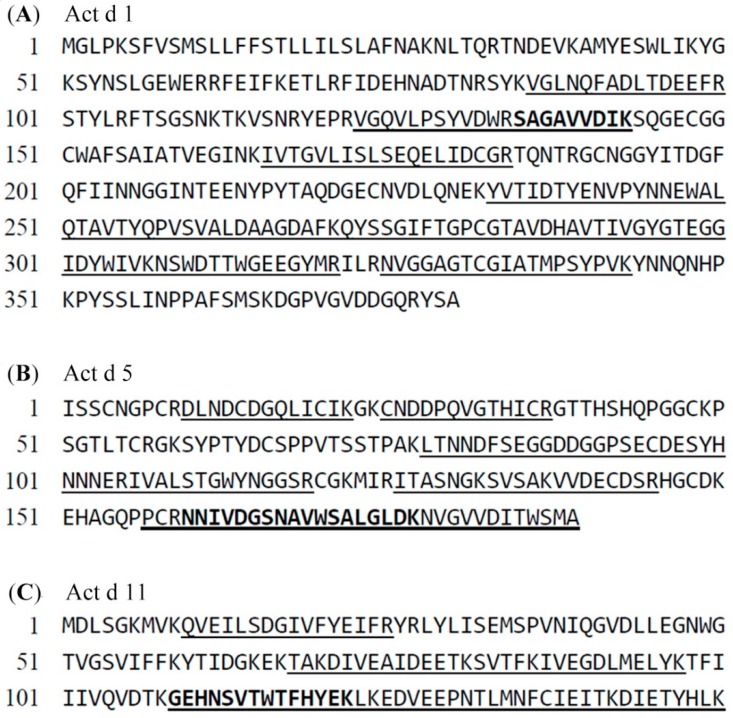
Protein sequences of (**A**) actinidin (Act d 1), (**B**) kiwellin (Act d 5), and (**C**) kirola (Act d 11). The tryptic peptides identified with LC/ESI-Q-ToF are underlined, and the selected signature peptides are presented in boldface type.

**Figure 4 molecules-24-01920-f004:**
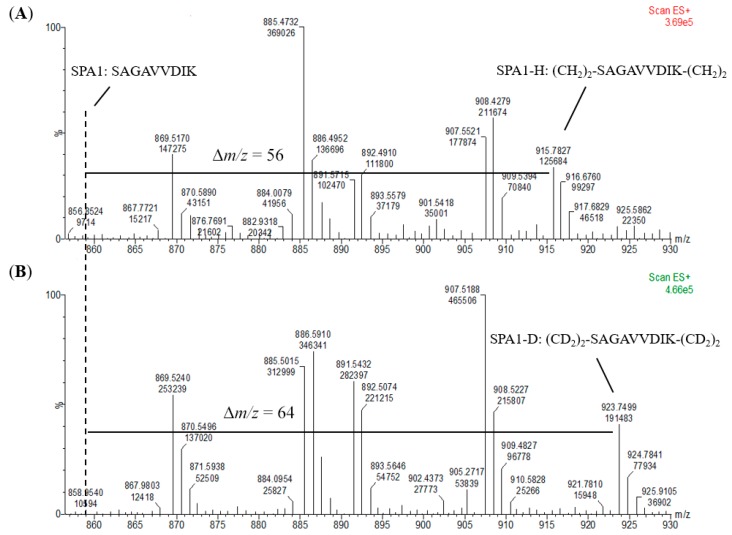
Mass spectra of (**A**) SPA1-H and (**B**) SPA1-D. The dotted line represents the *m*/*z* of SPA1 and Δ*m*/*z* shows the *m*/*z* shift of the signature peptide before and after stable-isotope dimethyl labeling.

**Figure 5 molecules-24-01920-f005:**
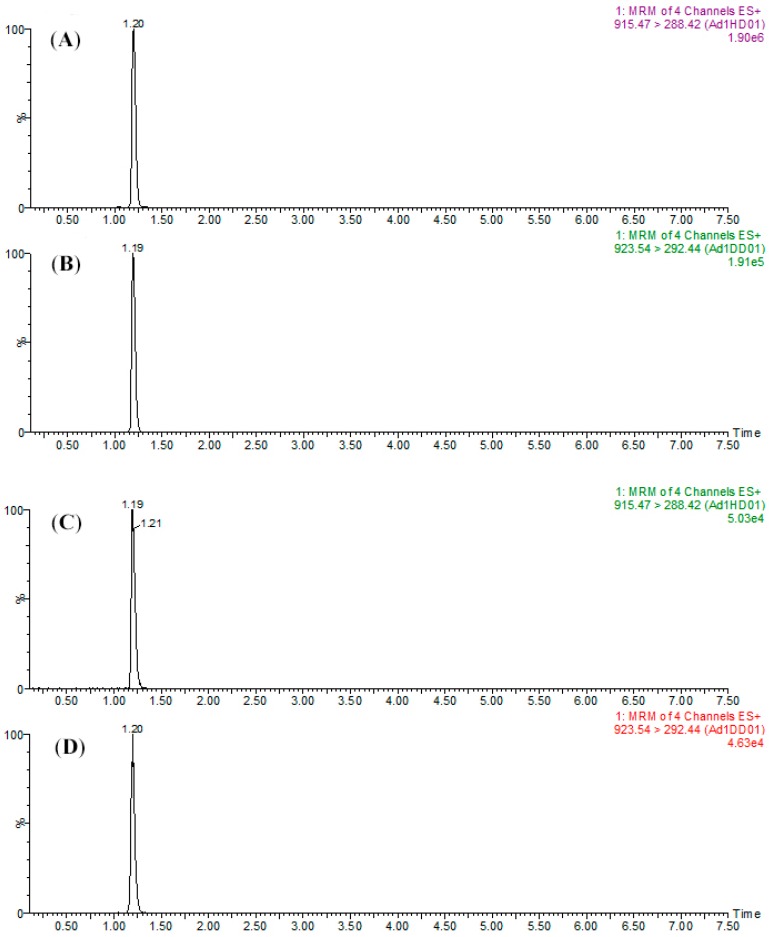
MRM chromatograms of (**A**) SPA1-H (1 μg mL^−1^) and (**B**) SPA1-D (500 ng mL^−1^) for the mixture of peptide standards and (**C**) SPA1-H (479.3 ng mL^−1^) and (**D**) SPA1-D (500 ng mL^−1^) for kiwifruit raw extract samples.

**Figure 6 molecules-24-01920-f006:**
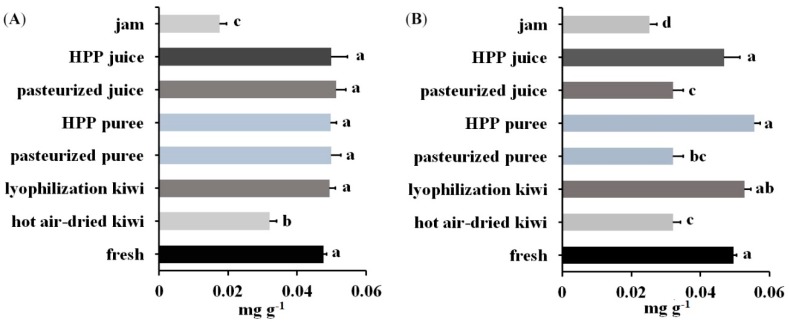
Content of (**A**) Act d 1 and (**B**) Act d 5 in one gram of kiwi foods produced by different processing. There was no significant difference between treatments with the same letter (*p*-value < 0.05. *n* = 3).

**Table 1 molecules-24-01920-t001:** The parameters of LC-MS-MRM analysis for dimethyl-labeled peptides.

Peptide Code	Stable Isotope Dimethyl Labeled Peptide Sequence	Retention Time (min)	Precursor Ion (*m*/*z*)	Z ^a^	Cone Voltage (V)	Product Ion (*m*/*z*)	Collision Energy (V)
SPA1-H	(CH_2_)_2_-SAGAVVDIK-(CH_2_)_2_	1.2	915	+1	55	288	45
						628	40
SPA1-D	(CD_2_)_2_-SAGAVVDIK-(CD_2_)_2_	1.2	923	+1	55	292	40
						632	40
SPA5-H	(CH_2_)_2_- NNIVDGSNAVWSALGLDK-(CH_2_)_2_	2.9	965	+2	65	460	30
						584	25
SPA5-D	(CD_2_)_2_- NNIVDGSNAVWSALGLDK-(CD_2_)_2_	2.9	969	+2	65	921	40
						1563	35
SPA11-H	(CH_2_)_2_- GEHNSVTWTFHYEK-(CH_2_)_2_	1.5	598	+3	50	466	25
						302	30
SPA11-D	(CD_2_)_2_- GEHNSVTWTFHYEK-(CD_2_)_2_	1.5	600	+3	45	527	20
						855	20

^a^ the charge state of precursor ion.

**Table 2 molecules-24-01920-t002:** Coefficient of determination, limit of detection (LOD), limit of quantification (LOQ), recovery, and precision of the target peptides.

Peptide Code	*R* ^2^	LOD ^a^	LOQ ^a^	Spike ^b^	Recovery (%)	Precision (% CV)
SPA1-H	0.9990	0.008	0.008	12500	109.9	1.86
				1562.5	83.9	5.11
				1250	78.9	9.92
				1000	75.6	8.51
				125	62.5	7.67
SPA5-H	0.9936	4	8	25000	75.8	3.44
				3215	86.3	9.55
				2500	84.9	9.37
				2000	72.5	3.87
				250	97.8	17.2
SAP11-H	0.9908	8	20	1250	39.8	11.7
				156.25	— ^c^	—
				125	68.7	36.4
				100	341.9	72.3
				12.5	—	—

^a^ ng mL^−1^, ^b^ ng, ^c^ cannot be determined.

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
