# Peer review of "Feasibility of Utilizing Stable-Isotope Dimethyl Labeling in Liquid Chromatography–Tandem Mass Spectrometry-Based Determination for Food Allergens—Case of Kiwifruit"

_molecules, 2019, doi:10.3390/molecules24101920_

Reviewer 1 Report

The manuscript “Feasibility of Utilizing Stable-Isotope Dimethyl Labeling in Liquid Chromatography-Tandem Mass Spectrometry-Based Determination for Food Allergens—Case of Kiwifruit” by Yi-Chen Shih and collaborators describes the setup of a procedure that can be exploited for identification and quantification of fruit allergens in raw and treated foods. The method is based on the use of liquid chromatography-tandem mass spectrometry and signature synthetic peptides with stable-isotope dimethyl labeling. This method has been applied to (and validated for) the identification and quantification of three kiwifruit allergens (Act d 1, Act d 5 and Act d 11) in different food matrices.

GENERAL COMMENTS.

This study is well performed and clearly explained. The methods are appropiate for the objective of the study and the conclusions are supported by the results.

The procedure set to perform this study is of general value and can be adopted for identification and quantification of other food allergens.

SOME SPECIFIC COMMENTS.

1.Pag 3, Results, Paragraph 2.1 and Legend to Figure 2.

The use of words “precipitation” and “extraction” is confusing. It is not clear when the procedure is used for protein extraction or protein concentration. Usually the precipitation is used to concentrate the proteins, rather than to extract them. Please, re-write the Paragraph making clear the procedures used to extract the proteins and the procedures used to concentrate them.  

2.Legend to Figure 6.

“….produce…..” Is it for “produced”?

3. Pag. 11 and 12, Materials and Methods. “Homogenized kiwi powder”, “kiwi puree” and “kiwi pulp”.

Please give details on the fruit parts used in these experiments. Were the little amounts of kiwifruit used in these experiments representative of green fruit pulp,white fruit pulp and seeds?

Author Response

Responses to the comments from reviewer 1

Point 1:

Pag 3, Results, Paragraph 2.1 and Legend to Figure 2.

The use of words “precipitation” and “extraction” is confusing. It is not clear when the procedure is used for protein extraction or protein concentration. Usually the precipitation is used to concentrate the proteins, rather than to extract them. Please, re-write the Paragraph making clear the procedures used to extract the proteins and the procedures used to concentrate them.  

Response 1:

We had revised the use of words “extraction” and “precipitation” in paragraph 2.1 and the legend to Figure 2 (Page 3, Line 150-167, and Page 4, Line 169-173) to make it clear in the revised manuscript.

 Point 2:

Legend to Figure 6.

“….produce…..” Is it for “produced”?

Response 2:

We had changed “produce” to “produced” in the legend to Figure 6 (Page 10, Line 260) in the revised manuscript.

 Point 3:

Pag. 11 and 12, Materials and Methods. “Homogenized kiwi powder”, “kiwi puree” and “kiwi pulp”.

Please give details on the fruit parts used in these experiments. Were the little amounts of kiwifruit used in these experiments representative of green fruit pulp,white fruit pulp and seeds?

Response 3:

The fruit used in these experiments was first peeled and then homogenized with the blender before sample pretreatment.  Therefore, after the homogenization, the kiwi samples used in the experiments were mixtures, which contained the green fruit pulp, white fruit pulp and seeds.  This information had been provided in Page 12, Line 370, and Page 13, Line 381-400 in the revised manuscript.

Reviewer 2 Report

Please improve introduction section both with methods for determination of allergens in kiwi as well as about the types of allergens in kiwi. There is a lot of papers published as a review papers. It would be nice to explain in detail why Authors developed this method and why it is important for kiwi. I found around 6-8 papers in both cases. Moreover, I cannot find any constructive discussion in the field of methods comparision and advantages of the new method in compalre to e.g. HPLC methods and other LC-MS. Please revise aims of the research because in current form there is a part of results.

Author Response

Responses to the comments from reviewer 2

Please improve introduction section both with methods for determination of allergens in kiwi as well as about the types of allergens in kiwi. There is a lot of papers published as a review papers. It would be nice to explain in detail why Authors developed this method and why it is important for kiwi. I found around 6-8 papers in both cases. Moreover, I cannot find any constructive discussion in the field of methods comparision and advantages of the new method in compalre to e.g. HPLC methods and other LC-MS. Please revise aims of the research because in current form there is a part of results.

Response:

We had described the comparison of method for allergen determination and the types of allergens in kiwi in the revised manuscript.

Page 1, Line 29-40 in the revised manuscript:

Sensitive and reliable methods for the determination of food allergens are necessary to detect the undeclared allergens resulting from contamination during food production and to enforce the regulation of allergen labeling [1].  For these purposes, various techniques have been applied to detect either the allergen protein itself or a marker indicating its presence [2].  Enzyme-linked immunosorbent assay (ELISA) is commonly used for allergen analysis.  However, ELISA is limited by the possibility of cross-reactivity which leads to false-positive results in this antibody-based assay.  DNA-based methods, which is carried out by means of detecting specific DNA fragments instead of the allergenic compounds, could not provide the exact concentration of the allergens [1].  Because of these limitations, alternative strategies for allergen analysis is required.  Mass spectrometry (MS)-based proteomic methods, providing high sensitivity and allowing the identification and quantification of allergenic protein, thus have been of great importance for allergen detection [3].

Page 3, Line 81-92 in the revised manuscript:

The quantification of Act d 1 and Act d 2 were carried out by means of ELISA and liquid chromatography-ultraviolet (LC-UV) method, respectively [25, 26].  However, these methods suffer from some disadvantages such as false-positive results caused by cross-reactivity in ELISA and overestimation of allergen contents resulted from the low selectivity in absorbance assay [27].  Allergen detection using liquid chromatography-tandem mass spectrometry in multiple reaction monitoring mode (LC-MS-MRM) is performed by analyzing the m/z values and ion transitions of signature peptides for target allergens, which could improve the selectivity and prevent the erroneous quantification [27].  Moreover, higher sensitivity could be achieved within MRM mode [5], indicating that the LC-MS-MRM could be an appropriate method for the analysis of kiwi allergens at trace level.

We also revise the aims of the research in the revised manuscript.

Page 3, Line 93-105 in the revised manuscript:

In this work, we intended to propose the first-time application of stable-isotope dimethyl labeling to the trace quantification of fruit allergens using LC-MS-MRM.  The identification of allergens was carried out with liquid chromatography/electrospray ionization-quadrupole-time of flight mass spectrometry (LC/ESI-Q-TOF).  The identified allergens with high coverage and MASCOT scores were selected as target allergens.  To develop the platform for the quantification of kiwi allergens, the external and internal standards were prepared by isotopically labeling the synthetic signature peptide standards with stable-isotope dimethyl labeling.  In addition, some steps for protein extraction described in a previous study [28] was optimized to be more time-efficient.  Moreover, the established procedure was validated with certain criteria and was applied to the self-made kiwi foods to examine the applicability for foodstuff analysis.

Reviewer 3 Report

The paper developed and validated a quantification procedure for food allergens using LS-MS-MS and dimethyl-labeled peptides as an internal and external standard. The signature peptides used as standard were first identified using preliminary tryptic digest and LC-MS analysis. Adequate recoveries for the selected standard peptides were obtained during spike addition. The method was successfully applied to the quantification of allergenic proteins in various kiwi food products. The paper is very interesting and is suitable for publication in Molecules. The following points are suggested to further clarify specific points in the paper.

1. The introduction can be improved by briefly discussing the traditional method for quantifying kiwi allergens (e.g. ELISA) and stating the advantages of LC-MS over traditional methods.

2. The amounts of proteins recovered in each protein extraction method can be stated in the results. 

3. Among the protein precipitation methods, TCA appears to be the most efficient based on the SDS-PAGE profile. Considering that the TCA method is also compatible with LC-MS analysis and does not involve salt, please discuss further why phenol extraction method was chosen over the TCA method. Will TCA be more effective in developing a general procedure for food allergen isolation and detection with LC-MS?

4. In figure 4, the font of the m/z value of the SPA1 peak can be enlarged for clarity.

5. In table 1, it may be useful to indicate the charge of the precursor ion in parenthesis since some product m/z ions are bigger than the precursor m/z ions.

6. In table 2, indicate the unit of the amount of spike added.

Author Response

Responses to the comments from reviewer 3

Point 1:

The introduction can be improved by briefly discussing the traditional method for quantifying kiwi allergens (e.g. ELISA) and stating the advantages of LC-MS over traditional methods.

Response 1:

We had provided further discussion about the traditional method for the quantification of kiwi allergens and stated the advantage of LC-MS-MRM method over the traditional methods in Page 1, Line 32-40, and Page 3, Line 79-92 in the revised manuscript.

 Point 2:

The amounts of proteins recovered in each protein extraction method can be stated in the results. 

Response 2:

We had stated the amounts of proteins recovered in the phenol and TCA methods (4.47 mg g–1 and 4.05 mg g–1, respectively) in Page 3, Line 117-118 in the revised manuscript.  The further investigation of the recovered amounts of proteins in other methods were not performed in our experiments.  This was because the poor efficiency of the other methods could be clearly determined with SDS-PAGE analysis.  Therefore, the exact amounts of proteins recovered by these methods were not investigated.

 Point 3:

Among the protein precipitation methods, TCA appears to be the most efficient based on the SDS-PAGE profile. Considering that the TCA method is also compatible with LC-MS analysis and does not involve salt, please discuss further why phenol extraction method was chosen over the TCA method. Will TCA be more effective in developing a general procedure for food allergen isolation and detection with LC-MS?

Response 3:

The phenol extraction method was proposed to be able to minimize the protein degradation within sample preparation steps [29] and thus could minimize the recovery loss to provide accurate quantification results.  Therefore, the phenol method was chosen over the TCA method for sample preparation.  This information is presented in Page 3, Line 120-125 in the revised manuscript.

 Point 4:

In figure 4, the font of the m/z value of the SPA1 peak can be enlarged for clarity.

Response 4:

We had replaced the original mass spectra with enlarged ones for clarity in Figure 4 (Page 6, Line 177).

 Point 5:

In table 1, it may be useful to indicate the charge of the precursor ion in parenthesis since some product m/z ions are bigger than the precursor m/z ions.

Response 5:

We had indicated the charge state of each precursor ion within Table 1 (Page 7, Line 196).

 Point 6:

In table 2, indicate the unit of the amount of spike added.

Response 6:

We had indicated the unit of the amount of spike added within Table 2 (Page 9, Line 210)